# The Use of Innovative Diagnostics to Inform Sustainable Control of Equine Helminth Infections

**DOI:** 10.3390/pathogens12101233

**Published:** 2023-10-11

**Authors:** Jacqueline B. Matthews, Natalia Peczak, Kirsty L. Lightbody

**Affiliations:** Austin Davis Biologics Ltd., Northamptonshire NN14 4BL, UK; natalia.peczak@austindavis.co.uk (N.P.); kirsty.lightbody@austindavis.co.uk (K.L.L.)

**Keywords:** horse, helminth, cyathostomins, *Anoplocephala perfoliata*, diagnostics

## Abstract

Helminths are commonly found in grazing equids, with cyathostomin nematodes and the cestode *Anoplocephala perfoliata* being the most prevalent. Most horses harbour low burdens of these parasites and do not develop signs of infection; however, in a small number of animals, high burdens can accumulate and cause disease. Cyathostomins are associated with a syndrome known as larval cyathostominosis. This occurs when large numbers of larvae emerge from the large intestinal wall. This disease has a case fatality rate of up to 50%. *A. perfoliata* infection has been associated with various types of colic, with burdens of >20 worms associated with pathogenicity. Anthelmintic resistance is a serious problem in cyathostomins and is emerging in *A. perfoliata.* Control methods that reduce reliance on anthelmintics now need to be applied, especially as no new dewormer compounds are on the horizon. Sustainable control methods must employ diagnostics to identify horses that require treatment. Coprological tests (faecal egg counts, FECs) have been used for several decades to inform treatment decisions to reduce helminth egg shedding. These tests cannot be used to assess host burdens as FECs do not correlate with cyathostomin or *A. perfoliata* burdens. In the last decade, new tests have become available that measure parasite-specific antibodies, the levels of which have been shown to correlate with parasite burden. These tests measure antigen-specific IgG(T) and are available in serum (cyathostomin, *A. perfoliata*) or saliva (*A. perfoliata*) formats. Tests for other helminths have been developed as research tools and need to be translated to support equine clinicians in practice. A key element of sustainable control strategies is that diagnostics must be used in combination with management approaches to reduce environmental transmission of helminths; this will help limit the proportion of horses harbouring parasite burdens that need to be targeted by treatment. This manuscript provides a review of the development, performance and general utility of various diagnostic methods for informing equine helminth management decisions.

## 1. Introduction

Horses are exposed to a range of gastrointestinal helminths. The most prevalent are the cyathostomins (small strongyles, small redworms) and the cestode, *A. perfoliata*. Most horses harbour these parasites without showing signs of disease. Other types of worms commonly encountered in horses are the ascarid, *Parascaris* spp. (in foals/yearlings) and *Oxyuris equi*, a pinworm that can persist in some populations. Historically, the nematode *Strongylus vulgaris* was considered a major threat because of its potential to cause non-strangulating infarction colic; this parasite is rarely found in populations given regular broad spectrum anthelmintic treatments.

As a result of their high prevalence and potential to cause disease and develop anthelmintic resistance, cyathostomins are considered the primary parasitic pathogens of horses [1]. This group of nematodes comprises 51 species [2], although typically horses are infected with only 5–10 common species, with low burdens of rarer species. Cyathostomins infect virtually all grazing horses with prevalence rates frequently approaching 100% [3]. Systematic reviews indicate that species prevalence and infection intensity patterns are similar across the world; the most common species are *Cylicocyclus nassatus*, *Cyathostomum catinatum* and *Cylicostephanus longibursatus* [3]. Similarities in surveys, spatially and temporally, demonstrate that the different species’ prevalence has not been altered considerably by extensive use of broad spectrum anthelmintics for >40 years (reviewed in [3]). Features most relevant to the diagnosis of cyathostomins are their species and life cycle complexity, the wide spectrum of parasite burdens that can develop in individuals and high levels of drug resistance [1,4]. Young (<5 years) and geriatric (>20 years) horses and animals with concurrent disease (for example, pituitary pars intermedia dysfunction) are more susceptible to high burdens that can have a clinical effect. Considerable burdens (>1 million in some cases) can accumulate in individuals not subjected to appropriate control measures; horses with such burdens can develop a colitis syndrome known as larval cyathostominosis [5].

Three cestode species infect horses: *A. perfoliata*, *Anoplocephala magna* and *Anoplocephaloides (Paranoplocephala) mamillana*, with *A. perfoliata* by far the most common [6]. An analysis of surveys published from 1977 to 2020, in which tapeworm burdens were enumerated postmortem [7,8,9,10,11,12,13,14,15,16,17,18,19,20,21,22,23,24,25,26,27], demonstrates a broad range in prevalence between regions and years, with prevalence rates ranging from 4% (in Poland [12]) to 81% (in Ireland [7]). Although infections with <100 tapeworms are observed most frequently (from 61% horses harbouring < 100 worms [22] to 93.5% horses harbouring < 100 worms [20]), the infection intensity in some individuals can reach >2000 worms [17,22]. Horses of all ages are susceptible to infection, including foals < 6 months [27]. Although there appears to be no obvious sex bias, some studies report higher infection intensities in mares compared to males [11,14]. Challenges associated with the diagnosis of *A. perfoliata* are the relatively low burdens associated with development of clinical disease and the poor sensitivity of coprological methods for detecting the presence of infection.

This manuscript provides a review of the development, performance and general utility of various diagnostic methods available that can inform equine helminth management decisions.

## 2. Life Cycles

Horses ingest cyathostomin third stage larvae (L3) when grazing (Figure 1). These penetrate the large intestinal mucosa/sub-mucosa, where they become encysted within host tissue. Mucosal L3, classified as early L3 (EL3), can persist for up to 2.5 years [28,29,30] before they grow to late L3 (LL3) then moult to fourth stage larvae (L4), which emerge from the intestinal wall. The mucosal phase can be short, especially in animals that lack immunity (i.e., younger animals), in which the prepatent period, as assessed by FEC analysis after experimental infection with a mixed species isolate, has been measured as 5–6 weeks [31]. Larval emergence occurs in phases. A moult to L5 follows emergence of L4 into the intestinal lumen. Here, L5 mature to adults, which mate and release eggs which are excreted in faeces. The number of each developmental stage can vary immensely between horses, even in those of similar age and grazing history [32]. In faeces, eggs hatch to release first stage larvae, which then undergo two moults to reach infective L3. Egg hatching and rate of L3 development is climate-dependent. The L3 retain the cuticle of the second larval stage as a protective layer and can survive, even in freezing conditions, from one year to the next. These stages migrate from faeces to surrounding vegetation in order to infect the next host.

The mucosal encysted larval stages are key to accurate diagnosis and effective treatment approaches. Their development has been proposed to be governed by several factors; immunity, age and cooler environmental conditions prior to infection are thought either individually or in combination to favour slower mucosal larval development with a consequent increase in the pre-patent period [33,34,35,36]. Immunity is initially directed at slowing the parasite mucosal phase, resulting in increased populations of encysted larvae towards the end of the grazing season [28]. In animals exposed to long-term repeated challenges, in addition to effects on larval development, immunity is thought to suppress egg release and eventually kill all stages of the life cycle [34,35,37]. Cyathostomins can survive on pasture or within their hosts for long periods; effective management of these parasites needs to take these features into consideration.

The body of *A. perfoliata* is composed of segments (proglottids) which each contain body systems and male and female reproductive organs. Mature tapeworms are differentiated into non-gravid adults (no eggs in terminal proglottids) and gravid adults (eggs in terminal proglottids) [38]. There is no opening to the uterus, so eggs are released sporadically when proglottids disintegrate, meaning that coproscopic detection of eggs is highly variable. In some studies, exclusively juvenile populations in individuals have accounted for up to 20% of infections [14,19,39]. Usually, mixed or adult-only populations account for most infections; however, in these reports, only a small percentage of adults (16–17%) were observed to be gravid and release eggs [18,21]. The life cycle is shown in Figure 2. 

Horses with significant access to pasture or that graze permanently on pasture have a higher risk of tapeworm infection [13,40]. Some studies indicate a higher prevalence or infection intensity in autumn and winter, likely due to the maturation of worm larvae (cysticercoids) ingested in infected oribatid mites during grazing [11,14,18,21,23,27].

## 3. Anthelmintics and Resistance

Three anthelmintic classes are licensed for treatment of cyathostomins: benzimidazoles (fenbendazole, oxibendazole), tetrahydropyrimidines (pyrantel salts) and macrocyclic lactones (ivermectin, moxidectin). Efficacy against different cyathostomin stages varies between and within classes and is affected by the dose rate and frequency [41]. Only fenbendazole (administered for five consecutive days) and moxidectin have been shown to demonstrate significant efficacy against EL3. Studies show variable effects of moxidectin against EL3, possibly reflecting experimental design differences [4]. Due to this variation, moxidectin only has a label claim for efficacy against EL3 in some territories. Only two anthelmintics have demonstrated high efficacy against *A. perfoliata*: the pyrazinoisoquinoline compound praziquantel, and the tetrahydropyrimidine pyrantel (at double the dose demonstrated as effective against nematodes).

The occurrence of anthelmintic resistance is frequent in cyathostomin populations. Resistance to fenbendazole and pyrantel salts, demonstrated as a lack of efficacy in reducing egg shedding 10–14 days after treatment, is commonly detected [1,4] and FEC reduction test studies recently indicated macrocyclic lactone resistance in cyathostomins [42,43]. More commonly reported is reduced effectiveness of macrocyclic lactones in suppressing cyathostomin egg shedding, with suppression measured for durations considerably shorter than those measured when these anthelmintics were first licensed [44]. It is extremely concerning that all currently available anthelmintics no longer demonstrate the level of efficacy originally reported against cyathostomins. Anthelmintic resistance in *A. perfoliata* had not been reported until a recent study in mares and yearlings in the US identified the potential lack of efficacy of pyrantel pamoate and praziquantel against this species [45]. Previously, there had been anecdotal reports of the reduced effectiveness of both active compounds. As no new anthelmintic compounds appear to be coming to market in the foreseeable future, it is imperative that approaches are employed that reduce reliance on anthelmintics.

## 4. Pathogenicity of Cyathostomins

The majority of cyathostomin-infected horses do not show disease but, in some animals, mucosal larvae accumulate in large numbers (100,000 s) and emerge simultaneously to cause larval cyathostominosis [46,47]. Clinical presentation varies, but cases commonly have diarrhoea, weight loss and peripheral oedema [48,49,50,51]. In Europe, larval cyathostominosis has a well-recognised seasonal occurrence (winter/spring) [47], usually occurring in horses < 5 years-old [52], but the disease can occur at any age, from 4 months [53] to geriatric animals [54]. The disease is most often observed in individuals, but outbreaks can occur [55,56]. Larval emergence causes severe damage to the intestinal wall and the disease has a case fatality rate of up to 50% [5]. The pathogenesis is poorly understood; factors involved in triggering mass emergence remain to be established. The disease coincides with the natural period of larval maturation and may only arise when the scale of emergence is sufficiently sizeable to disrupt gut function. Administration of adulticidal anthelmintics has been identified as a predisposing factor [52]. This may be due to treatment killing luminal stages and preventing negative feedback effects on encysted larvae.

Little is known about burdens associated with larval cyathostominosis as there are no studies describing cyathostomin counts in fatal cases. This is because enumeration of all stages is technically challenging [57]. One study indicated that in ponies administered high infective doses (3.15–3.9 M L3), the larval establishment rate varied considerably (0.94–39.7%), with only one animal developing larval cyathostominosis [29]. This highlights the difficulty of predicting disease risk, likely influenced by both parasite burden and individual host inflammatory/immune responses. In addition to larval cyathostominosis, these nematodes have been associated with non-strangulating infarction colic [50], caecocaecal intussusception [58], caecal tympany [51] and non-specific mild medical colic [59].

## 5. Pathogenicity of *A. perfoliata*

*A. perfoliata* attaches to the intestinal mucosa via head suckers which results in mucosal lesions, the severity of which has been observed to correlate with parasite burden. Most tapeworms attach to the caecal wall and ileocaecal junction; however, horses with higher burdens may also have tapeworms attached to the terminal ileum and ventral colon [15]. One study [10] identified that when low tapeworm numbers were present in the caecum, only 30% of lesions showed diphtheresis, but similarly, low tapeworm numbers at the ileocaecal junction resulted in 81% of worms being associated with diphtheritic lesions. When larger burdens were present (i.e., >21 tapeworms), all attachment sites were ulcerated with diphtheresis. Similarly, other studies [60] demonstrated that horse groups with 20 tapeworms or more had ulcerative lesions, severe, deep inflammation and oedema in the submucosa, whilst others [61] found a significantly higher burden of tapeworms (72–248 tapeworms) in groups with regional necrotising enteritis compared to groups categorised as having a lower range of burdens which had a less severe pathology. Similar observations were made in other studies [14], which demonstrate a relationship between the level of tapeworm infection in individuals and the severity of associated intestinal lesions.

Several studies demonstrate that the presence *A. perfoliata* infection is associated with the incidence of colic. For example, [62] used a centrifugation/flotation FEC method and demonstrated *A. perfoliata* as a causative agent of ileocaecal colic, reporting an incidence rate at 24 episodes/100 horses. In a follow-up matched case–control study [63], the risk of ileal impaction and spasmodic colic was assessed using host IgG(T) responses to *A. perfoliata* 12/13 kDa antigens as a measure of infection; these studies demonstrated a link between infection intensity and colic, with stronger correlations at higher levels of infection. Likewise, positive correlations between tapeworm egg shedding and serological positivity for *A. perfoliata* with various types of colic were identified [64]. A significant association between the presence of *A. perfoliata* eggs in faeces of horses with signs of colic was also observed in [65]. Other studies have failed to show a significant association between tapeworm infection, as detected by parasite specific serum IgG(T), and risk of colic [66,67]. This could be due to the broad scope of the colic definitions in these studies, due to a higher prevalence of *A. perfoliata* in test populations, or due to recent anthelmintic administration confounding the analysis due to the persistence of parasite-specific IgG(T) after treatment. The correlation of tapeworm egg detection with colic observed in some studies may be due to the fact that the FEC methods employed only detected eggs in horses with higher tapeworm burdens [14,68].

Given the capacity of cyathostomins and *A. perfoliata* to cause disease, it is important that high burdens are avoided and the need to treat infection must be balanced with a requirement to minimise selection pressure for anthelmintic resistance. This can be achieved by using diagnostic tests combined with improved pasture management to reduce parasite transmission. Full removal of faeces from paddocks at least once a week should be applied to break the life cycle of both parasites. This frequency of removal will reduce pasture contamination with infective stages, which, in the case of nematodes, are infective larvae or larvae within eggs, and in the case of cestodes, are infective cysticercoids within the intermediate oribatid mite host. Faeces should be removed well away from paddocks where horses graze and not near water courses due to the potential to contaminate the latter with anthelmintics (in particular, ivermectin) that may have been administered.

## 6. Faecal Egg Count Testing for Detecting Cyathostomin Infection

Testing is useful for monitoring nematode egg shedding levels to select horses for treatment to reduce contamination in pastures. As a result of the negative binomial distribution of cyathostomin infections (generally, <20% of an adult horse population excretes >80% of egg contamination), FEC-directed treatments can result in considerable reductions in anthelmintic use [69]. Compared to previously recommended all-group treatments, FEC-led protocols apply lower selection pressure for resistance as a proportion of the worm population in untreated horses are left unexposed to anthelmintics. Seasonality in cyathostomin egg shedding and the biology of free-living stages should be taken into account when applying FEC tests, with the higher egg shedding [70,71,72,73] and faster L3 development [74,75] from late spring to late summer making these periods the most applicable for FEC testing to inform treatment decisions. In northern temperate regions, cyathostomin FECs tend to be lower in autumn/winter due to encysted larvae accumulating in the caecum and colon wall [28,76]. At these times, risk assessments need to be applied with respect to encysted larval burdens to assess whether individuals require treatment with larvicidal anthelmintics to reduce disease risk and to mitigate increases in their FEC when larvae emerge and mature to adult worms.

FEC tests do not provide information on cyathostomin burdens—studies that compared strongyle egg counts to parasite counts at necropsy (the gold standard measure of parasite burden) showed no significant associations at higher egg shedding levels [32]—nor do FECs bear any relationship with larval burdens, which can comprise the majority of the burden [77]. Acquired immunity can limit egg production by female cyathostomins [37], further highlighting that FECs should not be used to estimate parasite burdens within hosts. This is important because horses exhibit considerable ranges in cyathostomin burdens; in a UK study [76], it was demonstrated that in 86 horses presenting at an abattoir, the luminal cyathostomin count range was 12,000–1,239,000. Likewise, in a necropsy study in the US [78] which surveyed 55 horses, the reported range of adult cyathostomins was 680–663,100.

As cyathostomin burdens do not correlate with FECs and encysted larvae increase at the end of the grazing season, experts previously recommended that all horses be treated in autumn/early winter with anthelmintics with licensed efficacy against all stages, including encysted larvae [79]. Due to extensive benzimidazole resistance, moxidectin was the most often anthelmintic recommended for this purpose [1]. Such all-horse treatments are likely to contribute to resistance; as indicated above, a risk assessment should be undertaken to assess if treatment is needed. The Small Redworm Blood Test (Guidelines for use of the test are available at Small Redworm Blood Test (austindavis.co.uk)) ELISA (Austin Davis Biologics Ltd., Northamptonshire, UK) can be used to help inform this treatment decision (see below).

FEC tests are useful for assessing anthelmintic efficacy by performing a FEC reduction test (FECRT) in which FECs are performed prior to, and 10–14 days after, treatment. For inclusion, horses should have a pre-treatment FEC of at least 200 eggs per gram (EPG) and a minimum of six horses should be assessed. When used for this purpose, a FEC method with a low multiplication factor should be applied [69]. FEC reduction thresholds for acceptable efficacy against cyathostomins have recently been updated [80]. If there is an insufficient mean reduction in FECs post-treatment (<90% or <95% depending on the anthelmintic administered), treatment failure or resistance is suspected.

## 7. Cyathostomin ELISA

This ELISA measures serum IgG(T) levels to three recombinant antigens representing *C. catinatum*, *C. nassatus* and *C. longibursatus*, recorded as the most prevalent species (88–93%) worldwide [3]. Initial development showed that, in experimentally infected ponies, serum IgG(T) to encysted larval antigens increased within 5 weeks post-infection and that antibody responses were largely directed against complexes of ~20 and ~25 kDa [81]. When these complexes were purified, serum IgG(T) levels specific to each were shown to correlate with the cyathostomin burden in infected animals [82]. To develop a test that would not need access to postmortem material as an antigen source, genes encoding immune-dominant antigens within the two complexes were identified [83]. Two genes were selected using cyathostomin-specific sera to screen an encysted larvae complementary (c)DNA library; these encode proteins, Gut Associated Larval Antigen [83] and Cyathostomin Diagnostic Antigen (CID) [84]. The transcript for GALA was found to be expressed in EL3 and late L3/developing L4 stages, whilst that coding for CID was detected in late mucosal larvae and luminal stage worms. Initially, 14 recombinant proteins representing GALA or CID proteins from nine cyathostomin species were studied to ascertain their individual and combined value in informing on parasite burden. The analysis indicated that a cocktail comprising three proteins (CT3) representing *C. nassatus*, *C. longibursatus* and *C. catinatum* provided excellent diagnostic performance without including additional proteins [84]. Analysis of CT3-specific IgG(T) in cyathostomin-infected versus uninfected horses showed that the Receiver Operator Characteristic (ROC) Area Under the Curve (AUC) values exceeded 0.9, demonstrating excellent test performance for detecting infection and diagnosing burdens above thresholds of up to 5000 cyathostomins [84]. The CT3 ELISA was subsequently optimised and validated in a commercial laboratory [85]. Optimisation included addition of equine IgG-based calibrators to generate standard curves to enable quantification of antigen-specific IgG(T) and provide integral QC for each sample. From the amount of CT3-specific IgG(T) quantified, an algorithm is applied to generate a ‘serum score’ for each horse. Validation of this optimised format was performed using cyathostomin-negative and -positive gold standards to assess test performance and to select serum score cut-off values for diagnosing burdens of up to 10,000 cyathostomins (Table 1).

To develop inclusion criteria for the test, Lightbody et al. [85] studied herds kept under different management/climactic conditions to assess the proportion of horses tested that fell above/below the 14.37 serum score cut-off for 1000 cyathostomins [85]. Strongyle FEC datasets were used to analyse concurrent/recent FEC patterns with CT3-specific IgG(T). This showed that application of this serum score cut-off would have led to a 41% reduction in anthelmintic treatments (296/719 horses below the cut-off) compared to an all-group treatment approach. The proportion of horses that fell below this serum score was associated with transmission factors such as FECs, with a significant difference in serum score results between FEC-positive and FEC-negative horses, with more FEC-negative horses below the 14.37 threshold (70%) compared to FEC-positive horses (24%) (Fisher’s exact test; *p* < 0.0001) and significantly more horses with FECs < 200 EPG below the 14.37 serum score cut-off (49.3%) compared to those with FECs ≥ 200 EPG (14.4%) (Fisher’s exact test; *p* < 0.0001) [85]. The results indicate the ELISA is best considered for informing treatment decisions when recent FECs in individuals and grazing companions are low (i.e., <200 EPG). In such cases, when selecting the burden threshold to apply for treatment (1000, 5000, 10,000 cyathostomins), an assessment of infection risk based on knowledge of the group (age, clinical condition) and management practices (stocking density, pasture hygiene) should be applied. Where transmission is judged to be high (high stocking density, no pasture hygiene measures, high proportions of young animals) and FEC results are consistently ≥200 EPG, the test is not recommended for informing treatment decisions, as many horses are likely to return a result above the serum score thresholds. Assessment may indicate that horses are at risk of harbouring pathogenic burdens that may require treatment to target all cyathostomin stages. Being serum-based, the test provides an opportunity for veterinarians to engage with clients in avoiding all-group treatments, especially where guidelines recommend that all horses be treated in autumn/winter or at the end of the grazing season with a ‘larvicidal’ anthelmintic [79,86].

Table 2 summarises factors to consider when using this test to provide information for anthelmintic treatment decisions.

After anthelmintic treatment, residual antibodies from past infection can have a confounding effect on test results. The serum half-life of equine IgG(T) has been reported to be between 21 [87] and 35 days [88]. To reduce the risk of false positives, it is recommended that the test is not applied until 4 months after treatment. Foal serum IgG(T) responses to cyathostomin infection occur within 6–12 weeks of birth [29]. After this, maternal antibodies from colostrum diminish, so animals over 3 months are considered appropriate for testing [85].

The test can be used by veterinarians in making a differential diagnosis in intestinal disease cases. There is no published research regarding the level of burden that causes larval cyathostominosis. Diagnosis is challenging and relies on exclusion of other conditions [89] and assessment of non-specific blood biochemical and haematological markers [47,49,51]. The test has been employed in larval cyathostominosis outbreaks [55] to indicate levels of parasite-specific serum IgG(T); here, all affected horses that were tested returned high serum scores (53.7–70.9). Only one case had an FEC > 200 EPG, highlighting a role for this ELISA in providing information in support of differential diagnosis in practice. This test has high sensitivity for detecting horses with negligible/low burdens and can differentiate horses in this group well from those with high burdens; thus, it has value as a ‘rule out’ test to exclude these parasites in the aetiology of other intestinal conditions.

## 8. Faecal Egg Count Testing for Detecting Tapeworm Infection

Coprological methods that apply flotation or sedimentation solutions have been used to diagnose *Anoplocephala* spp. infections. Studies that validated these FEC methods for detection of eggs against the gold standard of postmortem tapeworm counts indicate that although specificity is generally high (90–100%), sensitivity is low [14,15,26,68,90]. For example, conventional McMaster, sedimentation or flotation methods that use 2–8 g faeces had a sensitivity of 0–8%, 8% and 17–21%, respectively [16,22,26,91]. Modified McMaster, Cornell–Wisconsin and sedimentation/flotation methods demonstrated a sensitivity of up to 62% [13,14,15,16,19,20,22,23,68,90,91,92]. Although one of these studies [19] reported some correlation between *A. perfoliata* burden and number of eggs detected, the others revealed no significant relationship; all methods were unable to differentiate between horses with a few tapeworms from those with larger burdens, and false negative results were usually observed with burdens of <20 tapeworms. This is of relevance as several surveys that assessed tapeworm counts postmortem indicted that in 61–93.5% of horses, tapeworm infections comprised <100 parasites [13,14,19,20,68,90,91], highlighting a major limitation in using FEC analysis to identify tapeworm-infected horses.

A further important complicating factor in using coprological analysis for diagnosing tapeworm infection is the intermittent release of proglottids from adult worms, leading to temporally uneven egg excretion [6]. Individual proglottids have been shown to contain >1000 eggs [38]; therefore, if eggs are released together from a single proglottid and faeces are not well mixed, a sub-sample collected for analysis could result in a considerable over- or under-estimation of the egg count. Studies where faecal samples were spiked with tapeworm eggs indicated that standard coprological methods for detecting eggs provide high sensitivity [15,93], so it is likely that the consistently reported low sensitivity of FEC methods for detecting tapeworm infections is associated with the uneven release and distribution of eggs. An additional confounding factor in using FECs for assessing tapeworm infection is that immature worms can comprise a substantial proportion of the burden; these stages will not be detected by a test that measures eggs. This is demonstrated by the fact that, in infections where there are higher numbers of mature gravid parasites, a higher sensitivity has been reported [15,19,22,94]. For all of the reasons above, it is likely that *A. perfoliata* prevalence will be underestimated by FEC testing. Steps can be taken to increase the sensitivity of egg detection; however, these make the technique laborious and time-consuming [95] and will not overcome issues related to the presence of high proportions of immature worms or sterile adults and sporadic proglottid release.

## 9. Tapeworm Serum ELISA

Serological testing offers an alternative method for detecting *A. perfoliata* infection and allows for larger numbers of samples to be efficiently processed for healthcare monitoring [96]. Initial studies compared serum antibody responses to *A. perfoliata* scolex antigen [97], whole worm extracts (WWEs) and excretory/secretory (ES) antigens [98]. Although a positive correlation between antibody responses to crude scolex antigen and tapeworm infection was observed [97], the complex nature of the antigen, which comprises >14 proteins, will require further investigation to increase assay specificity. Molecules in WWEs demonstrated a lack of detectable serum antibody response, but proteins of 12 and 13 kDa present in worm ES fractions, but absent in WWEs, were consistently bound in immunoblot analyses by antibodies in tapeworm-positive sera, but not tapeworm-negative sera [98]. Detection of *A. perfoliata* infection by measuring serum IgG and IgG(T) levels to purified 12/13 kDa ES components reported sensitivities of 56% and 63%, respectively [99]. IgA and IgE were also assessed as marker isotypes; however, no difference in IgA response between infected and non-infected horses was observed, and measurement of parasite-specific IgE showed a low sensitivity (44%) compared to measurement of antigen-specific IgG(T) (78% sensitivity) [100]. Further studies focussed on assessing the value of serum IgG(T) responses to the 12/13 kDa ES antigens and reported sensitivities of 70–71% and specificities of 68–78% [19,20]. Bohorquez et al. [94] demonstrated antigenic cross reactivity between the 12/13 kDa proteins of *A. perfoliata* and antigens in *A. magna*: however, as *A. perfoliata* is considerably more prevalent and co-infections with both species are often observed [18,27,101], this was not considered an issue in applying the test in practice. Other studies reported no cross-reactivity of the *Anoplocephala* spp. 12/13 kDa ES antigens with those present in common equine nematode species, *A. mamillana* or equine bots [94,99].

An ELISA based on measuring serum IgG(T) to the 12/13 kDa ES antigens defined in [98] was shown to provide positive correlations (Spearman’s correlation, 0.63) between antigen-specific antibodies and infection intensity [99]. This test was commercialised in the UK >20 years ago by Diagnostic Ltd., who subsequently investigated the relationship between colic and *A. perfoliata* infection intensity [96] and demonstrated an increase in the odds ratio for the disease when sample results exceeded an optical density (OD) of 0.2 (defined as the cut-off for diagnosing infection). A subsequent study of 84 horses infected with *A. perfoliata* that compared postmortem worm counts with 12/13 kDa ES antigen-specific IgG(T) reported a significant relationship between antibody levels and infection intensity [19]. Despite a moderate correlation between ELISA OD values and tapeworm burdens, variability in antibody responses meant that although serum IgG(T) could be used to identify horses requiring anthelmintic treatment, antibody measurements could not be used to determine exact tapeworm burdens in individuals. In this study, 66% of horses with no visible parasites had OD values above the positive OD threshold; this may have been a result of the persistence of antibodies following anthelmintic treatment.

Subsequently, an ELISA that also measures serum IgG(T) to the 12/13 kDa ES antigens was marketed by Austin Davis Biologics Ltd. as the ‘Tapeworm Blood Test’ (Tapeworm Blood Test (austindavis.co.uk)). This test uses a direct ELISA format and incorporates a calibration curve to act as an internal quality control and generate a ‘serum score’ for each horse. This test has been validated using sera from naturally infected horses for which *A. perfoliata* burden data were available [24]. Using 1+ and 20+ worm burden thresholds, serum score cut-offs of 2.7 and 6.3, respectively, were selected based on optimal sensitivity (85–89%) and specificity (78–80%) values. Using the 1+ tapeworm cut-off value (2.7), no horses with a potentially pathogenic burden of >20 tapeworms were misdiagnosed [24]. The Spearman’s correlation between tapeworm burden and serum score was 0.78, an improvement on the previous serum IgG(T) ELISA. Serum score data are categorised as ‘low’, ‘borderline’ or ‘moderate/high’; treatment is advised for horses with results in the latter two categories.

Following anthelmintic treatment, studies [99] report a decline in anti-12/13 kDa specific serum IgG(T) in as short as 28 days; however, the length of time required to return to below the treatment threshold varies, especially in relation to the level of IgG(T) measured before treatment. Later studies also indicated that following treatment, a decline in anti-12/13 kDa IgG(T) could be detected within 28 days, but the majority of horses exhibited significant reductions within 12–18 weeks, with decreasing IgG(T) being observed up to six months post-treatment [96,102,103]. As anti-cestode anthelmintics have no persistent anti-parasitic effect, antigen-specific IgG(T) reductions post-treatment are likely to be affected by horses being reinfected if they return to contaminated pasture and by the presence of tapeworms not eliminated by treatment [102,104]. As the serum half-life of equine IgG(T) is reported to be 21 [87] and 35 days [88], it is important to obtain a full treatment history before testing. Application of the serum tapeworm ELISA is therefore recommended for a minimum of 4 months after treatment [24,103].

## 10. Tapeworm Saliva ELISA

A saliva test, EquiSal^®^ Tapeworm (EquiSal, Austin Davis Biologics Ltd.), based on the 12/13 kDa ES antigens described above, has been developed and was launched commercially in 2014. Saliva testing enables non-invasive sampling for assessing tapeworm infection. This test measures antigen-specific IgG(T) using a combination of integrated ELISAs to account for variation in saliva flow and diet. Saliva is collected by the veterinarian or horse owner using a saliva collection swab with an indicator zone that turns pink when the required volume is collected. The test comprises three formats: a 12/13 kDa antigen-specific assay, a non-specific binding assay and an assay that measures total IgG(T). The total IgG(T) and non-specific binding components control for the effect of variability within samples, especially the effect of saliva flow on antibody concentration. This test was validated by comparing antigen-specific IgG(T) levels in saliva with *A. perfoliata* counts in 104 horses [24]. Similar to the serum ELISA, the validation study applied 1+ and 20+ burden thresholds to select saliva score cut-off values (−0.09 and 0.62 for 1+ and 20+ tapeworm burdens, respectively) based on optimal sensitivity (83–86%) and specificity (79–85%) values. The Spearman rank coefficient for the integrated three-ELISA saliva format was 0.74, demonstrating a positive correlation between tapeworm count and saliva score. The test categorises results as ‘low’, ‘borderline’ and ‘moderate/high’ and has been demonstrated to accurately identify all *A. perfoliata*-infected horses with a clinically relevant burden (>20 tapeworms) [24].

Antigen-specific IgG(T) decreases more rapidly in saliva than in serum. One study indicated that, in >70% of praziquantel-treated horses, parasite-specific IgG(T) in saliva decreased to below anthelmintic treatment threshold levels within 5 weeks [24]. In subsequent studies, when saliva samples were collected every 2–3 weeks post-treatment from 15 horses without access to grazing or kept on paddocks where faeces were fully removed daily, saliva scores in all horses fell below the ELISA treatment threshold by 12 weeks after treatment [105]. As treated horses can become reinfected rapidly when grazing on contaminated pasture, retesting after 12 weeks can indicate whether or not horses are exposed to ongoing tapeworm transmission.

Both commercial ELISAs have been used to examine tapeworm prevalence in naturally infected populations. For example, one study [106] compared tapeworm FEC results with data from the serum and saliva tests to assess infection prevalence in 48 farms in Germany. Cestode eggs were detected in 6.3% samples, compared to a 52.1% prevalence (serum) and a 75.7% prevalence (saliva) detected by ELISA, with a moderate correlation between results from the serum and saliva tests. The difference in prevalence reported by the ELISAs is likely due to the saliva method detecting lower and earlier infections as reported previously [24]. The serum ELISA is based on a systemic antibody response with persisting antibodies that may remain for up to 6 months [24], whereas mucosal antibodies, such as those detected in saliva, are likely to be produced and secreted at the site of infection [100]. These antibodies may be secreted through mucosal epithelial cells by transcytosis with the equine analogue of the neonatal Fc receptor, rather than the polymeric Ig receptor [24].

Table 3 summarises factors to consider when using the tapeworm ELISA tests to provide information for treatment decisions.

## 11. Testing for *Strongylus vulgaris* Infection

This parasite was previously considered a major threat to equine health and was prevalent at high levels before the introduction of macrocyclic lactones [107,108]. Since the 1980s, its prevalence has fallen considerably and anthelmintic resistance has not been reported; however, a Swedish study in 2017 indicated that, where anthelmintics had only been available under veterinary prescription based on a diagnosis of infection for 10 years, there was a three-fold increase in *S. vulgaris* prevalence compared to the level reported a decade before the change in prescribing regulations [109]. It should be noted that there was no association identified between *S. vulgaris* prevalence and colic incidence; nevertheless, the results indicate that in circumstances where there are considerable reductions in anthelmintic administration, surveillance is desirable. A PCR-based assay of *S. vulgaris* DNA in faeces has been developed [110], but as the test detects eggs in faeces and the prepatent period of this nematode is 6/7 months, this method is not ideal for surveillance purposes. An ELISA based on detection of the serum IgG(T) response to a recombinant antigen, SvSXP, has been developed [111]. This ELISA is reported to have a 73.3% sensitivity and an 81% specificity, and its application in research surveys in Europe and the US has identified higher prevalence rates (59.4–75.9%) than reported by coprological analyses of larvae in faeces [112]. The ELISA requires optimisation in regard to its specificity before it can be deployed in surveillance programmes. Until then, testing for this parasite relies on coprological assessment. Large strongyle eggs cannot be differentiated from cyathostomin eggs, so faecal culture is required to obtain L3 which can be differentiated based on the larval intestinal cell morphology.

## 12. Conclusions

To address the need for accurate tools that support evidence-based helminth control, antibody-based tests have been developed for cyathostomins and *A. perfoliata*. These have been used in the UK and Europe to support veterinarians in making treatment decisions where FEC testing is of limited value. Use of these tests has led to substantial reductions in anthelmintic applications compared to levels used in traditional interval treatment approaches. Such reductions should result in lower selection pressure for anthelmintic resistance, thus prolonging the longevity of these important medicines. Similar tests need to be made available for pathogens such as *S. vulgaris* to enable monitoring of the impact of reduced anthelmintic use to avoid undesirable sequelae.

## Figures and Tables

**Figure 1 pathogens-12-01233-f001:**
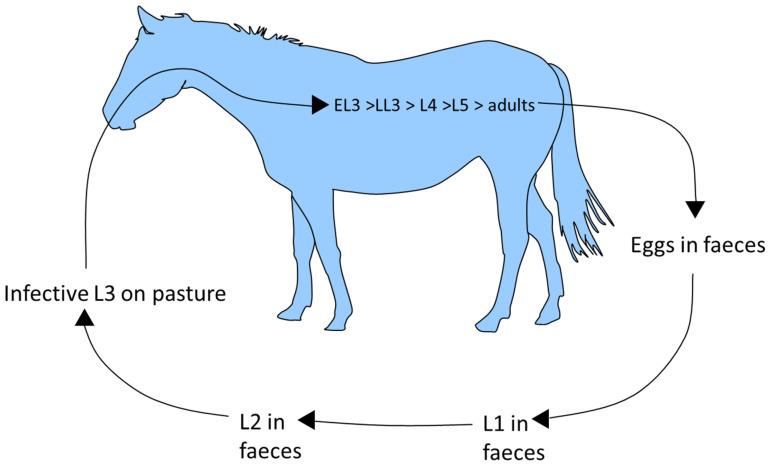
Cyathostomin life cycle.

**Figure 2 pathogens-12-01233-f002:**
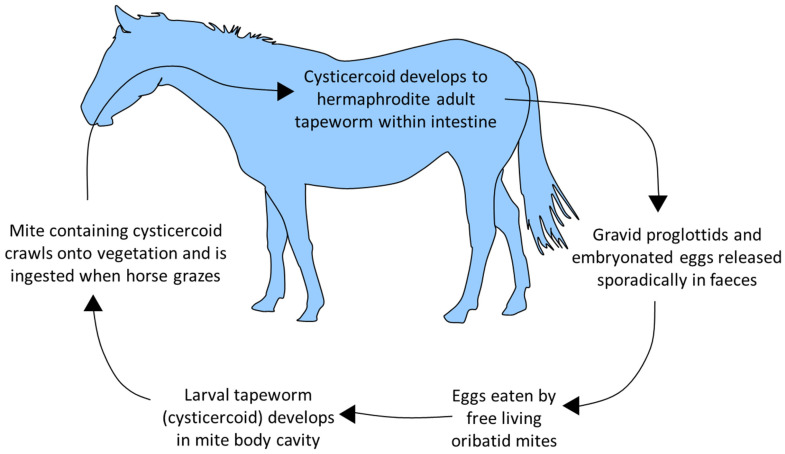
*A. perfoliata* life cycle.

**Table 1 pathogens-12-01233-t001:** Performance parameters for selected serum score cut-off thresholds in the Small Redworm Blood Test; 95% confidence intervals (CI) for each test performance parameter are included. Adapted from [85].

Parameter	Serum Score Threshold for >1000 Cyathostomins: 14.37	Serum Score Threshold for >5000 Cyathostomins: 15.61	Serum Score Threshold for >10,000 Cyathostomins: 30.46
Sensitivity (95% CI)	97.65% (91.76–99.71%)	96.10% (89.03–99.19%)	91.55% (82.51–96.84%)
Specificity (95% CI)	85.19% (66.27–95.81%)	71.43% (53.70–85.36%)	75.61% (59.70–87.64%)
Positive Likelihood Ratio (95% CI)	6.59 (2.67–16.29)	3.36 (1.99–5.69)	3.75 (2.18–6.46)
Negative Likelihood Ratio (95% CI)	0.03 (0.01–0.11)	0.05 (0.02–0.17)	0.11 (0.05–0.25)
Positive Predictive Value (95% CI)	95.40% (88.64–98.73%)	88.10% (79.19–94.14%)	86.67% (76.84–93.42%)
Negative Predictive Value (95% CI)	92.00% (73.97–99.02%)	89.29% (71.77–93.67%)	83.78% (67.99–93.81%)

**Table 2 pathogens-12-01233-t002:** Guidelines for assessing individuals for the Small Redworm Blood Test as an aid to informing anthelmintic treatment decisions.

	Low Infection Risk	High Infection Risk
^1^ Management factors to consider	^2^ Closed herd, dung removed >2 times per week, low stocking density (<2 horses/acre), no young stock (<5 years old) Horse has very limited access to pasture (e.g., sport or race horse)	^2^ Open herd, dung not removed/ removed sporadically, high stocking density (>2 horses/acre), high proportion of young stock (<5 years old), anthelmintic resistance reported
Recent faecal egg count results to consider	Concurrent/recent individual or group FEC results all <200 EPG	Individual or high proportion of group FEC results usually ≥200 EPG
Apply test?	YES	NO

^1^ Note that the individual conditions listed here can determine the decision towards applying the test or not, rather than the combination of the factors listed all being necessary to justify application of the test. ^2^ ‘Closed’ refers to a herd where new members are never or infrequently introduced. ‘Open’ refers to a herd where new members are frequently introduced or leave.

**Table 3 pathogens-12-01233-t003:** Guidelines for assessing individuals for tapeworm ELISA testing as an aid to informing anthelmintic treatment decisions.

	Low Infection Risk	High Infection Risk
^1^ Management factors to consider	^2^ Closed herd, dung removed >2 times per week, low stocking density (<2 horses/acre), no young stock (<5 years old), no previous history of colic in individuals or the group. Horse has limited access to pasture (e.g., sport or race horse)	^2^ Open herd, dung not removed/removed sporadically, high stocking density (>2 horses/acre), high proportion of young stock (<5 years old) present, previous history of colic reported in individuals or the grazing group, anthelmintic resistance reported
Previous tapeworm ELISA results to consider	Previous individual or all group tapeworm ELISA results reported in the low category	Individual or high proportion (>50%) of tapeworm ELISA results reported in the moderate/high category
Apply test?	YES—apply ELISA test once a year in either spring or autumn to inform need to treat with an anti-cestode anthelmintic product. Saliva test can be applied a minimum of 12 weeks after the previous anti-cestode treatment. Serum test can be applied a minimum of 4 months after the previous anti-cestode treatment	YES—apply ELISA test twice a year in spring and autumn to inform need to treat with an anti-cestode anthelmintic product. Saliva test can be applied a minimum of 12 weeks after the previous anti-cestode treatment. Serum test can be applied a minimum of 4 months after the previous anti-cestode treatment

^1^ Note that the individual conditions listed can determine the decision towards applying the tests or not, rather than the combination of the factors listed all being necessary to justify application of the tests. ^2^ ‘Closed’ refers to a herd where new members are never or infrequently introduced. ‘Open’ refers to a herd where new members are frequently introduced or leave.

## Data Availability

Not applicable.

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
