# Peer review of "The Use of Innovative Diagnostics to Inform Sustainable Control of Equine Helminth Infections"

_pathogens, 2023, doi:10.3390/pathogens12101233_

Round 1

Reviewer 1 Report

Minor editing of English language required.

Author Response

Please see our point by point response below. We thank for reviewer for their constructive critique of the manuscript. All changes have been underlined in the revised manuscript.

General comments and suggestions:
Helminths are commonly found in grazing equids, with the cyathostomin nematodes and the cestode. This review described the diagnostic tools for helminths in equids which addressed the need for evidence-based helminths control. The manuscript is scientifically sound and the methods are relatively explained. While the writing should improve. The comments are as follows.
Specific comments:
1. There were too many references cited in this manuscript which can be appropriately reduced to 60-80 references.

Author response; as this was a detailed review in a subject area that had not been published before, we do not consider it necessary to reduce the references, especially as the other reviewer did not suggest a reduction in references.  

2. Introduction.
(1) Line 41-42. May introduce “5-10 common species” in detail will make readers 
had better understanding.

Author response; apologies, we do not understand what this reviewer is suggesting here - is it to add species names?

(2) Latin names should be in full name when appear first, and the following should  be in abbreviation like “Anoplocephala perfolliata” in line 32 and line 58.

Author response; this has been changed in the revised manuscript. 

3. Life cycles.
(1) What’s the “LL3” in Figure 1. refer to? Please supply in the manuscript.

Author response; this has been clarified in the revised manuscript. 

(2) Please adjust the unsuitable triangle position in Figure 2.

Author response; this has been addressed in the revised manuscript. 

4. Line 182. Check “this results”.

Author response; this has been clarified in the revised manuscript. 

5. The diagnosis section ranged 6 to 10 is a bit messy overall. In “6 Diagnostic tests for cyathostomins”, only one subtitle is shown in this part and that is not necessary. It is the same in section 8. The sixth headline is “Diagnostic tests for cyathostomins” and that of eighth is “Diagnostics for tapeworm infection”. Then which worm is diagnosed for in the seventh paragraph? Or whether you need to rewrite the content division between the paragraphs? Subtitle exists in “9. Antibody based tests for diagnosing tapeworm infection” and “11. Testing for other parasites of concern” without number. In addition, there is an inconsistency in front size in the manuscript. 

Author response; we have revised the headings so that each section is more consistently described throughout the manuscript. All font sizes have been changed to 12, except the text within the tables which are included at font size 11. 

6. Tables in the manuscript should be in three-line

Author response; this has been changed in the revised manuscript. 

Reviewer 2 Report

General comments:

This is a useful review of the current evidence available for comparing various diagnostic methods for strongyle and tapeworm infection in horses in their utility for management decisions. I especially appreciated the combination of test results with environmental and demographic conditions in their guide for management decisions. I have only a few ‘major’ comments, and I have provided suggestions for improving the clarity and flow of the piece under minor comments.

Major Comments

-Lines 309-335: I like the nuance added in this paragraph, including the many factors to consider when weighing treatment options. However, it is unclear whether these analyses are included in Lightbody et al. 2023 or whether they are new analyses of these data presented here. Please make this clear and indicate what test was applied when reporting p-values. In addition, it would be much more useful to present a table with different epg cut-off values and their associations with serum score. I work with wild equids so my perspective is different, but to me 200epg seems like a very, very low egg count for triggering treatment.

-Lines 244-245: Nonlinear correlation is not uninformative if you know the parameters. Nielsen et al.’s findings were not that FECs are uninformative entirely, but rather that there is a significant association between FEC and cyathostomin burdens only up to an FEC of 500epg, above which there was no association. This is an important distinction and should be detailed here. It would also be useful to note that this study compared FECs to adult counts from necropsies (the gold standard), just like you do in the Anoplocephala section.

-I like Table 2 a lot! Could you please create a similar one for Anoplocephala? In addition, breaking up the lists in each cell into bullet points and titling the first row ‘Factors to Consider for Management’ would make it clearer that any individual conditions listed can push the decision towards applying the test or not, rather than the combination of the factors listed being necessary to justify the test. Also, please define ‘closed’ and ‘open’ herds somewhere, either within the table, in a footnote to the table, or in the text.

Minor Comments

-Please state the purpose of this piece in both the abstract and the introduction. Something like ‘Here, we review the accuracy, specificity, and general utility of various diagnostic methods for informing parasite management decisions’ would suffice.

-Line 35: there should not be a comma after ‘the nematode’ – otherwise it sounds like Strongylus vulgaris is THE nematode in horses. Same thing in line 283 after ‘these encode the proteins’.

-Line 42: delete ‘all’ and replace with ‘typically’

-Lines 47-49: please cite some of the surveys referred to.

-Line 73: ‘no obvious sex bias’, not ‘no sex obvious bias’.

-Lines 49-52 and lines 74-77: ‘Features most relevant to the diagnosis of…’ is very vague wording, making it unclear what you mean. Perhaps ‘Challenges associated with the diagnosis of…’ is closer?

-Lines 80-81: ‘within the host tissue.’ not ‘within a host tissue reaction.’

-Line 85: for which cyathostomin species was the prepatent period measured at 5-6 weeks? This ranges widely between species so you need to be specific here.

-Lines 89-92: it would be helpful to mention that the L3 stage is the stage that migrates from the faeces to surrounding vegetation in order to infect the next host, just to close the narrative loop in this paragraph.

-Fig. 1: Could you make it more evident that the L1 and L2 stages are both still in the faeces?

-Lines 96-99: more specific wording needed – do stronger immunity, intermediate age, and cooler environmental conditions favour encystment? How can these conditions favour both EL3 development AND a longer prepatent period? Did you mean slower EL3 development? Or faster EL3 development but longer L4/L5 stages?

-Line 101: ‘long-term’ should be hyphenated as it acts as one adjective. Same with ‘free-living’ in Fig. 2 and ‘follow-up’ in line 202, ‘Antibody-based’ in line 406 and in line 555, and ‘antigen-specific IgG(T)’ in line 488.

-Line 118: ‘horses with high levels of pasture’ is strange wording. 

-Line 154: ‘…emerge simultaneously to cause…’

-Lines 167-169: Very interesting!

-Line 191: ‘groups with 20 tapeworms or more’ is much more concise than ‘groups as having 20-100 tapeworms or >100 tapeworms…’

-Line 193: ‘found a significantly higher burden’ is fine

-Lines 200-202: careful here. Demonstrating causality is very difficult and typically requires experiment. What you describe here sounds like a correlation/association and I would word it as such.

-Line 215: Is it also possible that IgG(T) persists in circulation for quite awhile after infection? If so, this may be another reason for the lack of an association with colic.

-Line 249: you mean hosts exhibit considerable ranges in cyathostomin burden between individuals

-Lines 291-295: This is great!

-Lines 313-314: ‘a reduction to 41%’ not ‘a reduction of 41%’. Going from 719 treated horses to 296 is a 59% reduction.

-Line 353: Delete ‘here’, which makes it sound like you’re referring to the present study.

-Lines 355-357: I would say that the test is highly sensitive for detecting low burdens AND can differentiate these well from high burdens, therefore providing utility as a ‘rule out’ test.

-Lines 383-384: you mean ‘temporally uneven egg excretion’

-Lines 404-405: match font size with the rest of the paragraph. 

-Line 419: ‘led’ not ‘lead’.

-Line 493: ‘control for’ not ‘negate’

-Line 509: ‘horses’ not ‘horse’

-Line 512: delete the comma breaking up the clause ‘after which horses can rapidly become reinfected’.

-Line 513: ‘time period’ not ‘time scale’

-Line 537: a three-fold increase in ‘prevalence’, not ‘infection’

-Line 540: delete comma after ‘indicate that’

-Line 544: you mean a serum response to a recombinant antigen?

The quality of the English in this paper is great. Please see the 'Minor Comments' section provided for suggestions on improving clarity and flow in the writing.

Author Response

Point by Point response

Please note that all changes have been underlined in revised manuscript.

Major Comments

-Lines 309-335: I like the nuance added in this paragraph, including the many factors to consider when weighing treatment options. However, it is unclear whether these analyses are included in Lightbody et al. 2023 or whether they are new analyses of these data presented here. Please make this clear and indicate what test was applied when reporting p-values. In addition, it would be much more useful to present a table with different epg cut-off values and their associations with serum score. I work with wild equids so my perspective is different, but to me 200epg seems like a very, very low egg count for triggering treatment.

Author response; it has been clarified in the revised manuscript that the results are summarized from Lightbody et al. 2023 and the statistical test details are now included. Figures representing the table suggested here are presented in the Lightbody et al 2023 paper which is cited and so it is not felt necessary to repeat the information in this text. It is not the FEC that triggers treatment - FEC levels inform whether or not the test should be applied to inform on the need to treat in horses in low parasite transmission settings. 

-Lines 244-245: Nonlinear correlation is not uninformative if you know the parameters. Nielsen et al.’s findings were not that FECs are uninformative entirely, but rather that there is a significant association between FEC and cyathostomin burdens only up to an FEC of 500epg, above which there was no association. This is an important distinction and should be detailed here. It would also be useful to note that this study compared FECs to adult counts from necropsies (the gold standard), just like you do in the Anoplocephala section.

Author response; This has been clarified at this point in the text and reference to gold standard as per the later section is now included.

-I like Table 2 a lot! Could you please create a similar one for Anoplocephala? In addition, breaking up the lists in each cell into bullet points and titling the first row ‘Factors to Consider for Management’ would make it clearer that any individual conditions listed can push the decision towards applying the test or not, rather than the combination of the factors listed being necessary to justify the test. Also, please define ‘closed’ and ‘open’ herds somewhere, either within the table, in a footnote to the table, or in the text.

Author response; Glad you like the table - we have now added a Table 3 for A. perfoliata testing and have defined that a combination of factors should inform test deployment and have defined closed and open herds in footnotes to both tables. 

Minor Comments

-Please state the purpose of this piece in both the abstract and the introduction. Something like ‘Here, we review the accuracy, specificity, and general utility of various diagnostic methods for informing parasite management decisions’ would suffice. 

Author response; addressed in the revised text as recommended by the reviewer. 

-Line 35: there should not be a comma after ‘the nematode’ – otherwise it sounds like Strongylus vulgaris is THE nematode in horses. Same thing in line 283 after ‘these encode the proteins’.

Author response; addressed in the revised text as recommended by the reviewer. 

-Line 42: delete ‘all’ and replace with ‘typically’

Author response; addressed in the revised text as recommended by the reviewer. 

-Lines 47-49: please cite some of the surveys referred to.

Author response; addressed in the revised text as recommended by the reviewer - the review which covers all surveys is now cited clearly at this point 

-Line 73: ‘no obvious sex bias’, not ‘no sex obvious bias’.

Author response; addressed in the revised text as recommended by the reviewer. 

-Lines 49-52 and lines 74-77: ‘Features most relevant to the diagnosis of…’ is very vague wording, making it unclear what you mean. Perhaps ‘Challenges associated with the diagnosis of…’ is closer?

Author response; addressed in the revised text as recommended by the reviewer. 

-Lines 80-81: ‘within the host tissue.’ not ‘within a host tissue reaction.’

Author response; addressed in the revised text as recommended by the reviewer. 

-Line 85: for which cyathostomin species was the prepatent period measured at 5-6 weeks? This ranges widely between species so you need to be specific here.

Author response; the paper that was cited only examined eggs and larvae harvested from them. These cannot be differentiated to individual cyathostomin species - this has been clarified in the text. 

-Lines 89-92: it would be helpful to mention that the L3 stage is the stage that migrates from the faeces to surrounding vegetation in order to infect the next host, just to close the narrative loop in this paragraph.

Author response; addressed in the revised text as recommended by the reviewer. 

-Fig. 1: Could you make it more evident that the L1 and L2 stages are both still in the faeces?

Author response; addressed in the revised text as recommended by the reviewer and a revised figure has been added. 

-Lines 96-99: more specific wording needed – do stronger immunity, intermediate age, and cooler environmental conditions favour encystment? How can these conditions favour both EL3 development AND a longer prepatent period? Did you mean slower EL3 development? Or faster EL3 development but longer L4/L5 stages?

Author response; addressed in the revised text as recommended by the reviewer in order to clarify what is meant at this point in the review. 

-Line 101: ‘long-term’ should be hyphenated as it acts as one adjective. Same with ‘free-living’ in Fig. 2 and ‘follow-up’ in line 202, ‘Antibody-based’ in line 406 and in line 555, and ‘antigen-specific IgG(T)’ in line 488.

Author response; addressed in the revised text as recommended by the reviewer. 

-Line 118: ‘horses with high levels of pasture’ is strange wording. 

Author response; addressed in the revised text as recommended by the reviewer. 

-Line 154: ‘…emerge simultaneously to cause…’

Author response; addressed in the revised text as recommended by the reviewer. 

-Lines 167-169: Very interesting!

Author response; glad about that. 

-Line 191: ‘groups with 20 tapeworms or more’ is much more concise than ‘groups as having 20-100 tapeworms or >100 tapeworms…’

Author response; addressed in the revised text as recommended by the reviewer. 

-Line 193: ‘found a significantly higher burden’ is fine

Author response; addressed in the revised text as recommended by the reviewer. 

-Lines 200-202: careful here. Demonstrating causality is very difficult and typically requires experiment. What you describe here sounds like a correlation/association and I would word it as such.

Author response; addressed in the revised text as recommended by the reviewer. 

-Line 215: Is it also possible that IgG(T) persists in circulation for quite awhile after infection? If so, this may be another reason for the lack of an association with colic.

Author response; this has been clarified in the revised text as recommended by the reviewer. 

-Line 249: you mean hosts exhibit considerable ranges in cyathostomin burden between individuals

Author response; yes, this has been addressed in the revised text as recommended by the reviewer. 

-Lines 291-295: This is great!

Author response; :)

-Lines 313-314: ‘a reduction to 41%’ not ‘a reduction of 41%’. Going from 719 treated horses to 296 is a 59% reduction.

Author response; we have made this clearer - it was a 41% reduction in treatment recommendation based on this percentage of horses being under the 14.37 serum score in the ELISA. 

-Line 353: Delete ‘here’, which makes it sound like you’re referring to the present study.

Author response; addressed in the revised text as recommended by the reviewer. 

-Lines 355-357: I would say that the test is highly sensitive for detecting low burdens AND can differentiate these well from high burdens, therefore providing utility as a ‘rule out’ test.

Author response; good idea - thanks - and now addressed in the revised text as recommended by the reviewer. 

-Lines 383-384: you mean ‘temporally uneven egg excretion’

Author response; addressed in the revised text as recommended by the reviewer. 

-Lines 404-405: match font size with the rest of the paragraph. 

Author response; addressed in the revised text as recommended by the reviewer. 

-Line 419: ‘led’ not ‘lead’.

Author response; addressed in the revised text as recommended by the reviewer. 

-Line 493: ‘control for’ not ‘negate’

Author response; addressed in the revised text as recommended by the reviewer. 

-Line 509: ‘horses’ not ‘horse’

Author response; addressed in the revised text as recommended by the reviewer. 

-Line 512: delete the comma breaking up the clause ‘after which horses can rapidly become reinfected’.

Author response; addressed in the revised text as recommended by the reviewer. 

-Line 513: ‘time period’ not ‘time scale’

Author response; addressed in the revised text as recommended by the reviewer. 

-Line 537: a three-fold increase in ‘prevalence’, not ‘infection’

Author response; addressed in the revised text as recommended by the reviewer. 

-Line 540: delete comma after ‘indicate that’

Author response; addressed in the revised text as recommended by the reviewer. 

-Line 544: you mean a serum response to a recombinant antigen?

Author response; addressed in the revised text as recommended by the reviewer. 

Thanks for the constructive and detailed review!